# To Sample or to Specialise? Sport Participation Patterns of Youth Team Sport Male Players

**DOI:** 10.3390/children10040729

**Published:** 2023-04-14

**Authors:** Patrícia Coutinho, Ana Ramos, José Afonso, Cristiana Bessa, João Ribeiro, Keith Davids, António M. Fonseca, Isabel Mesquita

**Affiliations:** 1CIFI2D, Centre of Research, Education, Innovation and Intervention in Sport, Faculty of Sport, University of Porto, 4200-450 Porto, Portugal; 2Sport & Human Performance Research Group, Sheffield Hallam University, City Campus, Howard Street, Sheffield S1 1WB, UK

**Keywords:** athlete development, pathways, early diversification, early specialisation, practice, play, youth sports

## Abstract

This study characterised the sport participation patterns of 546 male youth team sport players. A retrospective questionnaire was used to identify the sport starting age (general sports and main sport) and the quantity and type of sports undertaken during the early years of development. A mixed-ANOVA and Chi-square tests were implemented. All participants started involvement in sports at the same age (~5 years) and participated in the same number of sports during their early years (1 to 2 sports). However, football players started participating mainly in team games (football, futsal) and water polo players in CGS sports (swimming). Participants reported different ages for initial participation in: (i) main sport (football players started participating earlier, around 5–6 years), (ii) onset of specialisation (football players specialised earlier, around 7 or 8 years), (iii) types of sports engaged in (football players were involved in more team games and water polo in more CGS sports), and (iv) variations in weekly training hours (water polo reported more hours of training). This study provided empirical evidence for understanding the effects of different sporting pathways on long-term athlete development. Some key incongruities between contemporary knowledge and practice are acknowledged. Further investigations should be developed by examining the trajectories in different sports, countries, genders, and cultural contexts.

## 1. Introduction

Early years of sport participation correspond to an important stage of long-term athlete development since they may define the trajectory of an athlete in sport, based on the quantity and type of practice undertaken [1]. Accordingly, two contrasting developmental pathways in sport have emerged in the literature and can be observed in practice: “early specialisation” and “early diversification” [2]. They differ in the exclusivity of focus in early, sport-specific practice (one sport vs. multiple sports), the quantity and quality of practice (proportion of structured/formal or unstructured/informal activities), and the level of engagement (usually expressed as number of hours of practice in a gross measure). Although they are often discussed within a dichotomic categorisation, they should be considered as two extremes of a possible continuum of other developmental pathways that are not yet known or (at least) defined in the literature.

Early specialisation involves participation in one sport, often to the exclusion of other sports, pursuing development in the chosen sport. Typically, this approach results in year-round participation and an early intensification of sport-specific, coach-led practice in a single sport, with the exclusion of peer-led play and involvement in other sports [3,4]. This approach to the development of expertise is exemplified by the deliberate practice approach, proposing a relationship between the quantity of specific, intense, and not inherently enjoyable, practice and performance [5,6]. Early specialisation is still a common pathway in sports where peak performance is achieved before adulthood/maturity (e.g., gymnastics, figure skating), and in some team sports like football (for a review, see [1]). Some researchers have suggested some negative long-term consequences associated with early specialisation such as physical and mental health problems including insufficient sleep, increased overuse injuries rates, overtraining, eating disorders, decreased sport enjoyment, depression, boredom, burnout, and dropout [1,7,8].

Despite such concerns, the prevalence of highly specialised youth athletes remains a concerning trend in the academic and practical domains of youth sport. For example, professional football clubs still invest time and money to recruit children as early as preschool ages (e.g., a 4-year-old ‘footballer’ scouted by Arsenal FC while still at nursery school in the UK: https://www.bbc.com/news/av/uk-58988254 (accessed on 21 October 2021)).

Alternatively, evidence continues to emerge demonstrating the benefits of an early diversification approach (reflected in the Developmental Model of Sport Participation [2,9]). This is characterised by participation in diverse youth sport activities in coach-led, structured, and organised practice as well as recommendations for greater involvement in child-led, unstructured, and informal activities [2,10]. Early diversification typically occurs during childhood, usually between 6 and 12 years of age (sampling years), before a gradual decrease in the number of activities undertaken and an ‘investment’ in one sport during mid to late adolescence [2].

The diversity of sporting experiences during the early years of development (6–12 years) occurs primarily for enjoyment and as a by-product, enriches functional athletic development, establishing the foundation for future participation in sport and further specialisation in a target sport [11,12]. Participating in a rich variety of sporting experiences (both formal and informal) allows children to experience diversified learning contexts, possibly leading to physical, cognitive, affective, and psychosocial enrichment [11]. This breadth and depth of experience is likely to enhance the intrinsic motivation that stems from the fun, enjoyment, and competence children experience in sport and physical activity [13,14]. Additionally, diversification may benefit skill performance as well as personal and social development (for a review, please see [12]).

Indeed, evidence continues to suggest that early diversification of sport experiences might promote skills transfer by exploiting affordance fields shared between sports and activities [13,14]. The Athletic Skills Model (ASM) [14,15], for example, is a pedagogical approach emanating from high performance sport, which is aligned with these ideas. Its principles encourage children to participate in multiple different sports to acquire relevant perceptual, cognitive, and movement competencies that can provide a powerful basis for later specialisation [15]. Many sports could be considered as “*donors*”, since they act as complementary enriching activities that may promote the transfer of global and specific skills and behaviours across a range of play and practice environments, supporting performance functionality at the specific moment of specialisation [15,16,17]. Important abilities that are critical to the development of an athlete can be “donated” by the participation in other sports that share adjacent fields of an affordance landscape, which can support skills transfer from a donor sport to a target sport [15,16,17].

The mix of research and theoretical understanding discussed so far suggests that the initial years of participation in play, physical activity, and sport are therefore an important phase in long-term athlete development. A diverse range of play and competitive experiences can support an athlete’s successful (or not) sport involvement as well as their motor, physical, psychological, and social development. Despite its relevance, little is known about how contemporary theories and knowledge may be filtering to the sporting pathways of youth team sport players, shaping the experiences of child participants. Such information is important to comprehend how practice in youth team sports is currently developing in accordance with contemporary knowledge and research on talent development regarding the need to engage in diversified sport practice and avoid the possible negative effects of early specialisation. This type of data analysis could provide important evidence to coaches, clubs, and sport systems to understand the effects of different sporting pathways on long-term athlete development in distinct sports from different countries and contexts.

In line with these ideas, the purpose of this study was to characterise the sport participation trajectories of male youth team sport players during the early years of sport development (6 to 12 years of age). Specifically, this study sought to characterise the sport starting age (of sports in general and in a main target sport) as well as the quantity and type of sport undertaken during this initial period of sport involvement.

## 2. Materials and Methods

### 2.1. Participants

The data for this paper came from the *In Search of Excellence in Sport–a mixed-longitudinal study in young male athletes* (INEX) research project carried out in Porto, Portugal, from 2017 through to 2019. The INEX study design is described in detail elsewhere [18]. The INEX study initially intended to analyse ~1000 players. Due to the dimension of the project and its characteristics, the final sample of the INEX study was ~600 players. The present study examined 546 youth male players from five team sports who demonstrated an availability to participate in this study. The sample included 166 basketball players, 113 football players, 111 handball players, 79 volleyball players, and 84 water polo players (average age of the overall players was 13.87 ± 1.9 years). These players came from 42 clubs affiliated with the Regional Sport Associations (basketball—13 clubs, football—8 clubs, handball—7 clubs, volleyball—6 clubs, water polo—5 clubs) and were selected to participate in this study by their coaches and/or club team managers since they had at least one year of specialised training and competitive experience in one of the team sports considered in this study. The five team sports (handball, basketball, football, volleyball, and water-polo) were considered since they were a convenience sample [18].

Ethical approval was granted by the Research Ethics Committee of the first author’s institution, guided by the Declaration of Helsinki (process number CEFADE 13.2017). Participants and parents were informed about the study’s purpose, their scope of involvement, and of their right to withdraw at any time. Data confidentiality was ensured, and informed consent forms were signed by both.

### 2.2. Data Collection

A retrospective questionnaire was designed to collect detailed information about the participants’ sport participation history including the quantity and type of sporting activities that they were involved in between the ages of 6 to 12 years. Three strategies were used to develop the questionnaire, fulfilling the requirements for validity and reliability. First, the underlying theoretical framework and a review of the literature examining similar related questions using available questionnaires was undertaken. This process contributed to item generation and the design of the first version of our questionnaire. Second, a panel of four experts in this research field evaluated whether the initial pool of questionnaire items represented the totality of the problems that may relate to the aims of this study. To estimate the content validity of the questionnaire, the panel of experts was asked to indicate whether each question was “essential” to measure the sport activities under analysis. The experts’ input relating to the assessment of each question was then used to measure the content validity ratio (CVR) [19]. The requirements of content validity were met since a value of 1.00 was reached in our analysis [19]. Finally, the revised version of the questionnaire was subjected to a pilot study with a sub-sample of 20 players to test the clarity and accuracy of the items, and the feasibility of the questionnaire. These data were not included in this study.

The initial part of the questionnaire was devoted to establishing a comprehensive set of variables related to the: (i) age of first sport participation, (ii) first sport participated in, (iii) age of first participation in the main sport, and (iv), age of specialisation in the main sport (i.e., the age when athletes start to be involved only in the target sport). The second part of the questionnaire recorded the range (number and type) of sports that the participants undertook during their initial stage of sport development (6 to 12 years of age) in an open answer question format. Moreover, the participants were asked about the amount of time (i.e., recorded as training hours) spent per week in those sport activities.

The participants’ reports referred to their involvement in formal organised sports (clubs/federations) only. Involvement in physical education classes or personal physical leisure was not considered. To analyse the type of sports participated in, a categorisation of such information was required. In line with some previous research [20], this study considered the categorisation of sports based on their competition rules, since they structure movement activity by defining the performance goal, providing modes of action and criteria for valuation. Therefore, the following types of formal competitive activities were considered: team games (e.g., badminton, volleyball, handball, field hockey, etc.), CGS (centimetres, grams, or seconds) sports (e.g., athletics, cycling, rowing, swimming, etc.), artistic composition sports (e.g., artistic gymnastics, figure skating, synchronised swimming, etc.), and combat sports (e.g., boxing, judo, taekwondo, karate, etc.) (for a complete definition of each categorisation, please see [21]).

For the data collection procedures, the participants recorded their retrospective reflections in writing, in a quiet classroom setting, under the supervision of one research team member. Due to the time constraints, the participants completed the questionnaire in a small group setting (six participants per group), although independently. After assuring the participants of the confidentiality and anonymity of their data, detailed verbal instructions were given. Explanations about the questions and the variables under analysis were also provided whenever needed. The questionnaire completion process lasted around 60 min.

### 2.3. Data Analysis

Descriptive statistics were computed to calculate frequencies, percentages, means, and standard deviation values. The normality and homogeneity of variance were checked through the Kolmogorov–Smirnov test and Levene’s test. A mixed-ANOVA was performed to test for statistical differences between groups in the following variables: sport starting age, age of first participation in the main sport, age of specialisation, number of sports practised (at all ages considered), and number of hours of training per week (at all ages considered). The effect size values were determined by the partial eta squared (*η*^2^_P_), considered as small (*η*^2^_P_ < 0.06), moderate (0.06 ≤ *η*^2^_P_ < 0.15), or large (*η*^2^_P_ ≥ 0.15) [22]. Post hoc analyses were conducted using Bonferroni tests (Bonferroni adjusted alpha of *p* = 0.001).

Chi-square was performed to test for statistical differences in participation frequencies between groups when considering the type of sports practised. The effect size values were calculated through Cramer’s V [23] and interpreted as a correlation [24] using arbitrary thresholds: very weak (0–0.19), weak (0.2–0.39), moderate (0.40–0.59), strong (0.6–0.79), and very strong (0.8–1) [25]. To assess the specific cells where differences emerged, adjusted standardised residuals (R) were calculated, with values ≥ |1.96|, implying that the cell had a number of cases significantly larger (or smaller, if negative) than expected [26]. Monte Carlo correction methods were used in the case where >20% of the cells had expected counts <5 [26]. An alpha level of *p* < 0.05 was considered statistically significant for all analyses. In addition to this quantitative data examination, a content analysis procedure was also undertaken to gain a better understanding of the type of sports practised when the previous quantitative analysis resulted in a statistically significant outcome.

To assess the reliability of the retrospective information provided by the participants, ~10% of the sample (i.e., 55 players—11 from each of the team sports considered) filled in the same questionnaire one month after the first moment of data collection. Pearson product-moment correlation values were calculated between the information collected at time one and time two. Correlation coefficient values were calculated as a function of the variables considered (i.e., sport starting age, age of first participation in the main sport, age of specialisation, number of sports practised, and hours of training per week from 6 to 12 years of age, and type of sports). The values were interpreted using the previously reported thresholds [25]. Reliability assessments showed strong correlation coefficients for all variables, with values ranging from *r* = 0.762 and *r* = 0.984. All of the reliability coefficients were statistically significant (*p* ≤ 0.001).

## 3. Results

### 3.1. Sport Starting Age, First Participation in the Main Sport, and Age of Specialisation

Descriptive and inferential statistics for sport starting age, age of first participation in the main sport, and age of specialisation in the main sport are presented in Table 1. Participants did not differ significantly in sport starting age, which occurred for all around 5–6 years of age. Notwithstanding, significant differences between participants were found concerning the type of sports practised in their first involvement in sport, although with weak correlations (χ^2^ = 103.48; *p* ≤ 0.001; V = 0.251). Football players started mainly participating in team games (adj.res. = 6.7), with qualitative content analysis showing that these were football and futsal activities. On the other hand, water polo players started sport involvement mainly with CGS sports (adj.res. = 7.9), with qualitative content analysis showing that these were swimming activities.

Concerning the age of initial participation in the main sport and age of specialisation, large significant differences were observed between participants. Football players started participating in football earlier (F_(4,541)_ = 42.647, *p* ≤ 0.001, *η*^2^_P_ = 0.241), around 5–6 years of age, while other participants start engaging in their main sports around 8–9 years of age. Football players specialised earlier in football (F_(4,543)_ = 15.257, *p* ≤ 0.001, *η*^2^_P_ = 0.102), which occurred around 7–8 years of age, whereas other participants specialised around 9–11 years of age.

### 3.2. Quantity and Type of Practice

Descriptive and inferential statistics for number of sports practised from 6 to 12 years of age are presented in Table 2. The participants did not differ significantly in the number of sports participated in during this period, with descriptive results showing that they were involved in around 1–2 sports in each year. However, significant differences were observed in the type of sports engaged in for football and water polo players. Football players were involved in more team games from the ages of 6 to 12 years and in fewer CGS sports from 6 to 11 years of age (Table 3). Qualitative content analysis for team games indicated that football players were involved in futsal activities at the ages of 6–7 years, and in football activities from 7–8 years onwards. Water polo players were more likely to be involved in CGS sports from 6 to 11 years of age and in fewer team games from 6 to 9 years (Table 3). Qualitative content analysis for CGS sports indicated participants were mainly involved in swimming activities until 11 years of age.

Descriptive and inferential statistics for hours of training per week from 6 to 12 years of age are presented in Table 4. Small but significant differences were found between participants when aged 10 years (F_(4,544)_ = 3.553, *p* = 0.007, *η*^2^_P_ = 0.026), and moderate differences were found for 11 years (F_(4,544)_ = 10.009, *p* < 0.001, *η*^2^_P_ = 0.069) and 12 years (F_(4,543)_ = 18.034, *p* < 0.001, *η*^2^_P_ = 0.118). Water polo players reported more hours of training compared to handball (*p* = 0.01) and football (*p* = 0.01) players when aged 10 years. Water polo players also reported more hours of training than all participants when aged 11 years (*p* < 0.001 for all) and 12 years (*p* < 0.001 for all). Football players reported fewer hours of training than basketball (*p* = 0.002) and volleyball (*p* = 0.005) players when aged 12 years.

## 4. Discussion

This study sought to provide evidence for a comparison of sport participation trajectories of male youth athletes from five different types of team sports (basketball, handball, football, volleyball, and water polo). The main aim was to provide data characterising the sport starting age in the sample (of sports in general and in the main sport), and the quantity and types of sports undertaken during the early years of development (i.e., 6 to 12 years of age). Globally, all participants started involvement in sports (in general) at the same age (around 5 years of age) and were involved in the same number of sports during their early sport development (around 1–2 sports). However, they reported different ages for initial participation in the main sport and the onset of specialisation as well as different types of sports engaged in and variations in the hours of training per week.

Initiation into sport occurred for all participants around 5–6 years of age, a finding that is in line with other data reported in existing studies on this topic [27,28,29]. These data showed that team sport athletes initiated their involvement in sport during the first stage of sporting development, normally situated approximately between 6 and 12 years of age [29,30,31]. Notwithstanding, differences between participants were observed concerning the age of initial participation in a main sport and age of specialisation. Here, large significant differences were observed for specific sports. Particularly, it was noted that football players started participating (around 5–6 years of age) and specialising (around 7–8 years of age) earlier in their main sport compared to the other youth team sport participants. While elite performance through early diversification has been observed in sports where peak performance is achieved in adulthood (e.g., in many team sports [1]), football seems to be an exception. Previous studies have shown that youth football players tend to start around 5–6 years of age in structured and intensive training in this sport [32,33].

While a relationship between quantity of specific practice and performance achievement in sport has been proposed in past research [34], early, single sport specialisation has been significantly associated with many challenges in future elite performance in team sports [1,8,12,35]. Evidence has highlighted the negative impact that early specialisation in sports may have for an athlete in a long-term perspective including insufficient sleep, increased overuse injury rates, emotional disturbances, overtraining, burnout, and eating disorders [7,36]. Empirical evidence supports these trends, with early specialisation being associated with an increased risk of overuse injuries, burnout, depression, and dropout [36,37].

On the other hand, basketball, handball, volleyball, and water polo participants reported a more diversified sport experience (involvement in 1–2 sports) before specialising in their main sport (which occurred around 10 years of age). Research suggests that diverse athletic exposure and sport sampling may enhance motor development and athletic capacity, reduce injury risk, and increase the opportunity for a child to discover the sport(s) or activities they will enjoy and possibly excel at [11,20,38]. Particularly in team sports, although there are elite players that specialised early in the target sport, evidence has also indicated that many other successful elite players participated in several sports before specialising in their main sport [27,28,39]. These ideas support theoretical suggestions about phases of talent development in young sport participants: one of the early enrichment of athletic capacities before the secondary specialisation period of dedicated practice in a target sport [13,15,40].

Despite the early diversified pathway identified in the cohort of youth team sport participants in this study, it is important to note that the specialisation age was earlier than that reported in previous studies. For example, the study of Coutinho and colleagues [30], with adult skilled and less skilled volleyball players, indicated that players, in general, specialised in their main sport around 12–15 years of age, although skilled players specialised later (14–15 years) compared to less skilled counterparts (12–13 years). Such findings are raising questions whether youth players nowadays may be specialising earlier than athletes did in the past. These findings need to be interpreted alongside concerns that children in contemporary generations are not as physically active as those in previous generations [14]. The lack of early functional movement experiences may act as a limitation and challenge for contemporary children on their trajectory developing into adult athletes.

Concerning the nature and type of sports practised, a clear pattern was identified for the football and water polo participants, but not for the other individuals (i.e., basketball, handball, and volleyball players). Football players tended to be involved in more team games (especially specialised football and futsal activities) until the age of 12 years, while water polo players were initially involved in CGS sports (swimming activities) until 11 years of age (specialisation age in water polo). Accordingly, team games and CGS sports could possibly act as complementary *donor sports*, in an early enrichment programme, since they may provide important experiences across a range of play and practice environments that support performance functionality at the moment of specialisation for football and water polo players [15,16,41]. These sports may share adjacent areas or fields of an *affordance landscape* [42] that include an extensive range of opportunities for action, which can support the transfer of functional performance behaviours to target sport participation [16]. Here, the transfer of learning could occur by using general movement behaviours, similar perceptual and contextual features, cognitive actions (i.e., problem-solving and decision-making under pressure) and physical conditioning capacities. These questions clearly raise the need for further empirical evidence and research to provide greater insights on this topic, with a particular interest in knowing if there are ‘better’ donor sports to enhance future abilities for specific sports or/and if there are ‘global’ donor sports that allow for the development of general abilities. Thus, further research could explore these issues and possibly try to understand the role of types of donor sports (for example, universal vs. specific donors).

The theoretical framework of ecological dynamics emphasises that talent development and learning in sport entails a smoothed transition between the generality and specificity of practice and transfer [40,43,44]. Richly varied play and practice experiences might potentiate exploratory and adaptive behaviours by inviting participants to continually satisfy different interacting constraints and educate their attentional focus and intentions during learning [44]. Such sporting experiences may have provided the participants in this study with the beginnings of a rich landscape of affordances that lead them to potentiate functional behavioural variability, and developing perceptual-motor exploration [13,45].

Additionally, participants in this study differed in the number of hours of training per week undertaken between 10 and 12 years of age. Water polo players reported undertaking more hours of training during this specific age period, which corresponded to the moment of specialisation (occurring around 10 years of age). This finding supports the theoretical tenets that the moment of specialisation corresponds to a period of significant investment in intensive and structured training, with an increased amount of specificity of practice [2,46]. However, caution is needed in interpreting this finding, since the specialisation of these participants is still emerging at a very young age. Huge amounts of specific practice may not be advantageous at this stage for an individual’s long-term development [47,48,49].

Indeed, a recent meta-analysis by Güllich and colleagues [50] revealed that higher performing *youth athletes* that started playing earlier in their target sport were involved in more specific practice and showed faster initial progress than youth athletes from lower performance levels. In contrast, the results showed that *adult world-class athletes* engaged in more childhood/adolescent multisport practice, started their main sport later, accumulated less main-sport practice, and initially progressed more slowly than national-class athletes. Such observations suggest that investment in high quantities of sport-specific practice likely increases the probability of early *junior* success, but compromises the sustainability of the long-term development of international senior success [50].

Despite the important findings of this study, there are some limitations that should be addressed. Although used widely in the literature, reliable and valid retrospective methodologies only reflect the interpretation of records and the participants’ reports/perceptions of their previous sport experiences. While these reflections are an important source of documented information, they do need to be triangulated with other objective data to provide a rich understanding of developmental patterns [51]. This was the same for the data science analyses of the participation rates and competitive outcomes analyses in junior and senior sports records [50]. Big data analyses need to be juxtaposed with more qualitative investigations to gain rich, perceptive analyses of expertise and talent development in sport [1]. Thus, contemporary research methods in sport science may need further evidence of *participant perception* on the nature, type, and influence of practice activities as well as the quantity of relevant units in their practice histories such as hours or the number of activities undertaken. When feasible, prospective longitudinal studies should be undertaken, although the required timescales (i.e., several years, maybe decades) make this methodology intrusive and difficult to implement.

The selected methodologies, therefore, cannot be based only on data mining since researchers have to guarantee that they are not disrupting, nor distorting the perceptions of the selected participants (whether coaches or athletes). Here, qualitative research methods (for example, in-depth interviews, focus groups, participant observation, reflective note taking, action research, ethnographic studies) may offer a deeper perspective to better understand the role of practice and play activities in athlete development and expertise achievement. Such methodologies may help researchers to better comprehend how training transfer facilitates athlete development. Further research on this topic should also explore other contexts that were not explored in this specific study. For example, the experiences of female athletes, athletes from other sports (e.g., other team sports, CGS sports, combat sports, artistic composition sports, etc.), sports from different level of popularity, and other cultural contexts (different countries and social communities, clubs from villages vs. clubs from big cities, different sport organisation such as school-model vs. club-model) need to be examined [12].

Moreover, the degree and extent of specialisation within a given sport may vary considerably and provide another highly relevant layer of analysis that should be considered in future research. Such evidence is needed to better understand the long-term athlete developmental processes and to better inform the professional work and ideology of coaches, sport systems, and organisations as well as talent development and identification programmes.

## 5. Conclusions

The findings of this study provided a brief characterisation of the sport participation patterns of male youth athletes from five different types of team sports (basketball, handball, football, volleyball, and water polo) in Portugal. Overall, the participants started their involvement in sports (in general) at the same age and participated in the same number of sports during their early sport development. However, they differed in the type of sports practised in their first involvement in sport, the ages for first participation in the main sport, and the onset of specialisation as well as the types of sports engaged in and variations in the hours of training per week. The results were clear in demonstrating that football players had a different sporting developmental pattern since they initiated their involvement in sport-specific formal training earlier. They might face pressures to specialise earlier in this sport by limiting their focus of experience, although they did not report more hours of training in their early years. Water polo players were also distinguished from other the participants when reporting their involvement in more CGS sports (swimming) from an early age. They reported more hours of training between 10 and 12 years compared to other participants.

Such empirical evidence provides a snapshot of how practice in youth team sports in Portugal is currently developing and allowed us to compare these practical approaches with contemporary scientific and theoretical insights. Additional research is needed to investigate the sporting developmental patterns in other types of sports and in other countries to understand how different social communities and cultures provide different sport development experiences for their children. This type of research would provide a consideration of features of best practice and process markers of athlete and talent development, leading to the emergence of robust guidelines to ensure children’s health and wellbeing as well as the implementation of successful long-term developmental pathways applied in practice.

## Figures and Tables

**Table 1 children-10-00729-t001:** Ages of first participation in sport (general and specific) and specialisation according to the sport.

	Basketball	Football	Handball	Volleyball	Water Polo	F (p*η*^2^)^2^	*p*
Sport starting age	5.47 ± 2.20	5.15 ± 1.74	5.28 ± 1.93	5.36 ± 2.40	5.05 ± 2.25	0.73 (0.01)	0.57
First participation in the main sport	8.56 ± 2.36	5.81 ± 1.98 *	9.11 ± 2.72	8.97 ± 3.18	9.83 ± 1.87	42.65 (0.24)	<0.001
Age of specialisation	9.64 ± 2.51	7.93 ± 3.11 *	10.04 ± 2.92	10.21 ± 2.86	10.60 ± 1.90	15.26 (0.10)	<0.001

* *p* < 0.05.

**Table 2 children-10-00729-t002:** Number of sports practised according to age and sport.

Age	Basketball	Football	Handball	Volleyball	Water Polo	F (p*η*^2^)^2^	*p*
6 years	1.16 ± 0.80	1.20 ± 0.76	1.31 ± 0.83	1.21 ± 0.93	1.10 ± 0.89	0.86 (0.01)	0.498
7 years	1.17 ± 0.78	1.20 ± 0.70	1.25 ± 0.72	1.21 ± 0.80	1.18 ± 0.76	0.18 (0.01)	0.949
8 years	1.16 ± 0.72	1.24 ± 0.62	1.12 ± 0.69	1.24 ± 0.72	1.07 ± 0.63	1.06 (0.01)	0.375
9 years	1.17 ± 0.70	1.21 ± 0.47	1.13 ± 0.73	1.17 ± 0.75	1.12 ± 0.60	0.31 (0.01)	0.873
10 years	1.14 ± 0.62	1.19 ± 0.44	1.13 ± 0.72	1.21 ± 0.73	1.11 ± 0.61	0.40 (0.01)	0.807
11 years	1.11 ± 0.52	1.12 ± 0.36	1.09 ± 0.52	1.18 ± 0.60	1.11 ± 0.52	0.35 (0.01)	0.847
12 years	1.07 ± 0.41	1.12 ± 0.35	1.12 ± 0.58	1.05 ± 0.39	1.15 ± 0.61	0.64 (0.01)	0.635

**Table 3 children-10-00729-t003:** Chi-square statistics (% and adjusted standardised residuals) for the type of sports practised according to age.

Age	Sport	Types of Sports								χ^2^	*p*	V
		Team Games		CGS sports		AC Sports		Combat Sports				
		%	adj.res.	%	adj.res.	%	adj.res.	%	adj.res.			
6	Basketball	29.7	−0.2	33.0	0.6	50.0	0.6	37.5	0.6	76.504	**<0.001**	0.187
	Football	34.5	**4.8 ***	5.0	**−4.3 ***	0.0	−0.7	18.8	−0.2			
	Handball	20.3	0.2	19.0	−0.2	0.0	−0.7	25.0	0.6			
	Volleyball	12.8	−0.6	10.0	−1.4	0.0	−0.6	18.8	0.5			
	Water Polo	2.7	**−4.9 ***	33.0	**5.6 ***	50.0	1.4	0.0	−1.7			
7	Basketball	31.5	0.4	28.0	−0.6	0.0	−0.7	38.5	0.6	86.737	**<0.001**	0.199
	Football	31.5	**4.9 ***	3.2	**−4.5 ***	0.0	−0.5	7.7	−1.2			
	Handball	18.2	−0.6	21.5	0.5	0.0	−0.5	15.4	−0.4			
	Volleyball	12.8	−0.8	8.6	−1.7	0.0	−0.4	23.1	0.9			
	Water Polo	5.9	**−4.6 ***	38.7	**7.0 ***	0.0	−0.4	15.4	0.0			
8	Basketball	30.3	−0.1	23.9	−1.5	0.0	−1.3	25.0	−0.4	80.906	**<0.001**	0.192
	Football	29.9	**4.7 ***	3.4	**−4.4 ***	0.0	−1.0	8.3	−1.1			
	Handball	17.8	−0.9	23.9	1.1	50.0	1.5	25.0	0.5			
	Volleyball	13.7	−0.4	11.4	−0.9	0.0	−0.8	16.7	0.2			
	Water Polo	8.3	**−3.9 ***	37.5	**6.4 ***	50.0	1.5	25.0	1.0			
9	Basketball	30.2	−0.1	22.8	−1.3	0.0	−1.3	18.2	−0.9	60.006	**<0.001**	0.166
	Football	27.8	**4.4 ***	3.5	**−3.4 ***	0.0	−1.0	9.1	−1.0			
	Handball	18.6	−0.7	26.3	1.4	50.0	1.5	36.4	1.4			
	Volleyball	12.0	−1.6	12.3	−0.5	0.0	−0.8	18.2	0.4			
	Water Polo	11.3	**−2.6 ***	35.1	**4.5 ***	50.0	1.2	18.2	0.3			
10	Basketball	29.9	−0.4	21.4	−1.1	20.0	−0.5	40.0	0.5	45.379	**<0.001**	0.144
	Football	25.2	**3.4 ***	3.4	**−2.8 ***	0.0	−1.1	0.0	−1.1			
	Handball	18.8	−0.6	32.1	1.7	20.0	0.0	20.0	1.3			
	Volleyball	12.8	−1.2	14.3	0.0	0.0	−0.9	0.0	−0.9			
	Water Polo	13.3	−1.5	32.1	**2.6 ***	20.0	1.2	0.0	−0.9			
11	Basketball	31.2	0.6	7.7	−1.6	0.0	−0.9	0.0	−1.3	45.726	**<0.001**	0.145
	Football	23.7	**2.9 ***	0.0	**−1.9 ***	0.0	−0.7	0.0	−1.0			
	Handball	18.5	−1.2	38.5	1.6	50.0	1.1	10.0	1.1			
	Volleyball	12.4	−1.6	15.4	0.1	0.0	−0.6	0.0	−0.8			
	Water Polo	14.1	−1.1	38.5	**2.4 ***	50.0	1.4	0.0	−0.8			
12	Basketball	31.3	1.2	0.0	−1.2	0.0	−0.9	26.8	−0.6	23.109	0.111	0.103
	Football	21.5	**2.0 ***	0.0	−1.5	0.0	−0.7	23.2	0.5			
	Handball	18.7	−1.3	22.2	0.2	0.0	−0.7	23.2	0.7			
	Volleyball	13.9	−0.6	24.4	1.2	50.0	1.4	10.7	−0.8			
	Water Polo	14.6	−0.7	33.3	1.5	50.0	1.4	16.1	0.2			

* *p* < 0.05 and adjusted standardised residuals (adj.res.) ≥ |1.96|.

**Table 4 children-10-00729-t004:** Number of hours of training per week according to age and sport.

Age	Basketball	Football	Handball	Volleyball	Water Polo	F (p*η*^2^) ^2^	*p*
6 years	3.41 ± 2.36	3.21 ± 2.51	3.49 ± 2.46	3.30 ± 2.62	2.84 ± 3.11	0.90 (0.01)	0.462
7 years	3.78 ± 2.38	3.44 ± 2.32	3.74 ± 2.40	3.58 ± 2.46	3.34 ± 3.05	0.62 (0.01)	0.646
8 years	4.17 ± 2.84	3.65 ± 2.05	3.67 ± 2.36	3.79 ± 2.25	3.86 ± 3.25	0.90 (0.01)	0.462
9 years	4.45 ± 2.46	3.90 ± 1.72	3.83 ± 2.34	3.92 ± 2.54	4.46 ± 3.38	1.78 (0.02)	0.131
10 years	4.61 ± 2.20	4.12 ± 1.67	4.11 ± 2.31	4.66 ± 2.58	**5.27 ± 3.60 ***	3.56 (0.03)	**0.007**
11 years	4.98 ± 2.10	4.26 ± 1.60	4.39 ± 2.19	4.86 ± 2.30	**6.25 ± 3.70 ***	10.01 (0.07)	**<0.001**
12 years	5.42 ± 1.94	**4.37 ± 1.56 ***	4.71 ± 1.90	5.53 ± 2.09	**6.98 ± 3.79 ***	18.03 (0.12)	**<0.001**

* *p* < 0.05.

## Data Availability

The data of this study are available from the corresponding author upon reasonable request.

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
