# Peer review of "To Sample or to Specialise? Sport Participation Patterns of Youth Team Sport Male Players"

_children, 2023, doi:10.3390/children10040729_

Round 1
Reviewer 1 Report
In my opinion, the authors did an excellent job demonstrating the sports participation paths of youth athletes in various team sports during the chronological age of 6 to 12 with a focus on the starting age and other related issues. In the Introduction, the authors "smoothly" concluded to the research question, leading the reader into a logical closing. At this point, a hypothesis could be formulated. The methodology is clearly explained, well written and easy to understand. The Discussion appears to be not solely a repetition of the findings already reported in the tables, but also included new insight and managed to explain what the findings mean.
Author Response
We really would like to thank the reviewer not only all the work developed in this revision, but also the kind words addressed to our study. In this specific, research area and in many scientific journals, hypothesis have been abandoned, since studies like the one we presented are not correlational neither purely explanatory. For that reason, hypothesis are not mandatory. We made a great effort to develop a clear research background and rational in order to allow the reader to understand the real purpose of our study. Anyway, reviewer#3 provided several comments and we updated our article according to his/her suggestions. Please, find attached the new version of our manuscript. Again, many thanks for all your contribution.

Reviewer 2 Report
First of all, I think that the topic is quite interesting for both the scientific and professional public. In this sense, it has great potential among the readership. In the introduction to the paper, the authors explain the concepts of sports specialization and diversification, which are key to understanding the importance of children's sports activities. The chosen statistical methods are adequate. The sample size is good, which enables quality data processing. The obtained results were correctly interpreted. The discussion was accompanied by relevant research, which was accompanied by a large number of references, some of which are quite recent (2022), which is certainly commendable. Conclusions were drawn in accordance with the obtained results. In any case, I have no objections and I believe that the paper should be accepted in its present form for publication.
Author Response
We would like to thank the reviewer for the kind words and all the revision work developed. We take the opportunity to inform that reviewer#3 provided several comments and we updated our manuscript accordingly. Please, find attached the new version of the manuscript. Again, many thanks for all your work.

Reviewer 3 Report
First of all, I want to thank to the authors their effort to broaden an on-trend, interesting topic. Basically, the paper is, in my opinion both well written and well structured. Nevertheless, there are some points that need further discussion and may elicit some changes. I hope the following remarks can help the authors to improve their manuscript.
Line 36 à although we can make the difference between formal/informal activities, this characteristic does not apply to the difference between specialization versus diversification i.e. playing football/soccer for a future basketball player at his home backyard with his siblings is informal, but it is not going to be characterized as diversification as the present study neither makes the difference; I would not mean that it is not an important factor, but informal physical activities are common in childhood and those who specialize earlier are likely to spend some time on them also, the difference when talking about free versus coach-led play comes more from the quantity or quality (i.e. goals, as pointed out in your reference [8]) than from their existence or not
Lines 71-2 à It is clear that sports sampling allows “experience diversified learning contexts, leading to physical, cognitive, affective, and psychosocial enrichment” but I am not sure about the last two factors (affective and psychosocial) since are not guaranteed just for the sampling per se.
Line 84 à Some sports can be donors but, again, this is not granted; related to that, I would like to read comments from the authors about the next questions:
- Are better sports acting as sample sports to enhance other sports future abilities? Is this dependent of the final sport played or some can be seen as universal donors?
- Some kids start to play in similar sports (i.e. football and basketball), are they in better position than those who have different experiences (i.e. judo and volleyball)? I mean, can be more transferred activities in one situation?
Lines 95-7 à I suggest to be extremely cautious with the statements, there is a lot of talent (if not most of them) that has been developed under these circumstances, especially when social and contextual factors are promoting it
Lines 111-117 à Is the sample proportional to the licenses from where it has been obtained? How were the coaches that selected the players selected? Why these sports have been selected?
Lines 137-9 à Was the pilot study results included in the final results? If so … have been any modification between the pilot and the final procedures?
Lines 192 à Why 10% of the overall sample has been selected to check reliability when sub-samples (each sport) are not the same?
Table 3 à The number of sports is a count so discrete data. I do not like to see it represented as means with decimals, a frequency table can be more appropriate (or percentages)
Discussion
I think that contextual factors can weight more than others when regarding the present topic. American model versus European model (or school-model versus club-model if you want) can provide some explanation or hypothese regarding these and other findings; I would add this as a limitation of the study (results are likely to be similar to other European countries than to US)
Lines 319-21 à Many successful players participated in several sports, and many do not, it seems to me bold. I do not know how much overlap both groups regarding to many of the outcomes studied (performance, injuries, burnout, …)
Lines 349-52 à How should be a physical conditioning at an early age if we are in a sampling environment? Sport-specific or common since some of the athletes will change of modality?
Author Response
We really would like to thank you the revision made. It was very insightful and fruitful for our manuscript.
Please, see below the answer to your comments:
Line 36 - although we can make the difference between formal/informal activities, this characteristic does not apply to the difference between specialization versus diversification i.e. playing football/soccer for a future basketball player at his home backyard with his siblings is informal, but it is not going to be characterized as diversification as the present study neither makes the difference; I would not mean that it is not an important factor, but informal physical activities are common in childhood and those who specialize earlier are likely to spend some time on them also, the difference when talking about free versus coach-led play comes more from the quantity or quality (i.e. goals, as pointed out in your reference [8]) than from their existence or not
Answer: We included this idea in the text.
Location: Page 1, Line 35-36.
Lines 71-2 - It is clear that sports sampling allows “experience diversified learning contexts, leading to physical, cognitive, affective, and psychosocial enrichment” but I am not sure about the last two factors (affective and psychosocial) since are not guaranteed just for the sampling per se.
Answer: Several research in the last 15-20 years have suggested and demonstrated that early sampling has the potential to provide benefits in several ways, namely: motor (physical, technical, tactical), cognitive (e.g., game understanding), and psychosocial (personal and social development). Regarding the psychosocial development, there are several reasons that can explain this: 1) sampling different sports offer distinct social contexts and opportunities for socialization, 2) sampling sports allow kids to be within different types of motivational climates and activities that promote intrinsic motivation; this will subsequently help children become more self-determined and committed in their future participation in sport, 3) sampling sports allow kid to be in contact with different coaches and peers/teammates, which therefore helps them to develop, for instance, leadership skills, auto-regulation, self-confidence, etc. All this is fully (and better) explained in the reference [12] and since this specific topic is not within the specific aim of the study, we decided to not develop it in the introduction and invite the reader to read more in the aforementioned reference (we include a sentence “for a review, please see [12]). Anyway, we carefully thought about your comment and decided to be more cautious when referring to these issues by including the word “possibly”.
Location: Page 2, Line 73.
Line 84 - Some sports can be donors but, again, this is not granted; related to that, I would like to read comments from the authors about the next questions:
- Are better sports acting as sample sports to enhance other sports future abilities? Is this dependent of the final sport played or some can be seen as universal donors?
- Some kids start to play in similar sports (i.e. football and basketball), are they in better position than those who have different experiences (i.e. judo and volleyball)? I mean, can be more transferred activities in one situation?
Answer: We really appreciated the opportunity to discuss about these issues in this review process. The questions are interesting and very important, and is something that researchers are trying to explore yet without a specific answer. The concept of ‘donor sports’ tell us that there are some sports that can ‘donate’ abilities to others. Although this is a somewhat hypothesised, only few studies explored this issues (references 15, 16 and 17 in our study). The truth is that research has not yet the possibility to really demonstrate this and to achieve the knowledge you are asking for in your questions. What research suggested ate this present time is that abilities deemed critical to athlete development can be “donated” by performance and experience in selected sports that share adjacent fields of an affordance landscape including an extensive range of opportunities for action that can support skills transfer from a donor sport to a target sport. We included this idea in the introduction. We also included the ideas behind your questions in the discussion section, especially as indications for further research.
Location: Page 2, Line 88-90, and Page 10, Line 363-369
Lines 95-7 - I suggest to be extremely cautious with the statements, there is a lot of talent (if not most of them) that has been developed under these circumstances, especially when social and contextual factors are promoting it
Answer: We reformulated the text accordingly.
Location: Page 2, Line 100.
Lines 111-117 - Is the sample proportional to the licenses from where it has been obtained? How were the coaches that selected the players selected? Why these sports have been selected?
Answer: The INEX study initially intended to analyse ~1000 players (200 from each team sports). Due to the dimension of the project and its characteristics, the final sample of the INEX study was ~600 players. The present study examined 546 youth male players from five team sports, which are the ones that demonstrated availability to participate in this study. These players came from clubs affiliated to the Regional Sport Associations and were selected to participate in this study by their coaches and/or club team managers since they had at least one year of specialised training and competitive experience in one of the team sports considered in this study. In other others, all players that comprise these characteristics and demonstrated availability to participate were included in this study. The five team sports (handball, basketball, football, volleyball and water-polo) since they are a convenience sample (Patton, 2002). This is explained in the reference [18]. All this information were included in the text.
Location: Page 3, Line 115-125.
Lines 137-9 - Was the pilot study results included in the final results? If so … have been any modification between the pilot and the final procedures?
Answer: The data from pilot study was not included in the present study. We included this information in the text.
Location: Page 4, Line 147-148.
Lines 192 - Why 10% of the overall sample has been selected to check reliability when sub-samples (each sport) are not the same?
Answer: We intentionally selected the same number of players from each team sports in order to have a balanced perspective of the reliability check data. The same procedures have been used in past research in this area (please, see, for example, reference 27, 28, 30).
Location: not included in the text.
Table 3 - The number of sports is a count so discrete data. I do not like to see it represented as means with decimals, a frequency table can be more appropriate (or percentages)
Answer: Although I completely understand your perspective (1 one sport is 1 sport and not 1,2 sports…), research in this area uses these variables as continuous. The same happens within Psychology research area, for example (Likert scales and its results). For that reason, and for having a comparative term in the future (for example, in a meta-analysis), we really would like to maintain the data as it is. Hopefully you can understand our pointview.
Location: none.
Discussion
I think that contextual factors can weight more than others when regarding the present topic. American model versus European model (or school-model versus club-model if you want) can provide some explanation or hypothese regarding these and other findings; I would add this as a limitation of the study (results are likely to be similar to other European countries than to US)
Answer: Your point of view is extremely valuable and correct. We included this in the end of the discussion section as a potential avenue to explore in further research.
Location: Page 11, Line 427.
Lines 319-21 - Many successful players participated in several sports, and many do not, it seems to me bold. I do not know how much overlap both groups regarding to many of the outcomes studied (performance, injuries, burnout, …)
Answer: We provided a more cautious perspective about these ideas in the text.
Location: Page 9, Line 330.
Lines 349-52 - How should be a physical conditioning at an early age if we are in a sampling environment? Sport-specific or common since some of the athletes will change of modality?
Answer: Why do we have to adopt an opposite perspective? Why can't we do a little bit of both? For sure is better than only one of them! That’s the notion of diversity/sampling and what we are trying to demonstrate in this study! :)
Location: throughout the discussion section.
Please, find attached the new version of the manuscript.
Again, many thanks for all your work!

Reviewer 4 Report
The paper is very interesting and well writen.
I have only a few comment to make about the article.
In methods two references are in other citation format (Cohen et al. and Irene et al).
Decimals are indicated in comma instead of a period in the sample description.
There could have been a bias when selecting the sample if they were taken from the same sport clubs. Could you include in the participants sections a detailed process of the sample recruitment (number of clubs, location), data no valid...? If the clubs come from village instead of cities the sport offer is more reduced so the children may have practiced only a few different sports. Could you include this information and consider its effect on the conclusions?
Was the age of starting competition measured? This should be a good data regarding the specialization age.
How the age of specialization was considered ? How it was explained in the questionnaire to the children?
Does the participants write the name of the sports practiced or should they select a category of sports?
Maybe Table 2 and 4 could be represented in a graph to see the evolution of the number of sports by age and the number of hours by age.
Beside the big limitation of considering only male athletes in the study, there is another one due to the fact of analysing in the study only mainstream team sports. Other team sports like rugby, hockey etc... due to its nature of minority sports could offer different data on specialization or the simultaneous practice with other sports during 6-12 ages.
Author Response
Please see the attachment (1 - response to reviewer#4 and 2 - manuscript)
